# Characterization and comparison of human and mouse milk cells

**Rose Doerfler[1]☯, Saigopalakrishna Yerneni[1]☯, Alexandra Newby[1], Namit Chaudhary[1], Ashley Shu[1], Katherine Fein[1], Juliana Hofstatter Azambuja[2], Kathryn A. Whitehead[1,3]***

**1** Department of Chemical Engineering, Carnegie Mellon University, Pittsburgh, PA, United States of America, **2** Department of Pediatrics, UPMC Children's Hospital of Pittsburgh, University of Pittsburgh, Pittsburgh, PA, United States of America, **3** Department of Biomedical Engineering, Carnegie Mellon University, Pittsburgh, PA, United States of America

☯ These authors contributed equally to this work.
* kawhite@cmu.edu

## Abstract

Recent data has characterized human milk cells with unprecedented detail and provided insight into cell populations. While such analysis of freshly expressed human milk has been possible, studies of cell functionality within the infant have been limited to animal models. One commonly used animal model for milk research is the mouse; however, limited data are available describing the composition of mouse milk. In particular, the maternal cells of mouse milk have not been previously characterized in detail, in part due to the difficulty in collecting sufficient volumes of mouse milk. In this study, we have established a method to collect high volumes of mouse milk, isolate cells, and compare the cell counts and types to human milk. Surprisingly, we found that mouse milk cell density is three orders of magnitude higher than human milk. The cell types present in the milk of mice and humans are similar, broadly consisting of mammary epithelial cells and immune cells. These results provide a basis of comparison for mouse and human milk cells and will inform the most appropriate uses of mouse models for the study of human phenomena.

## Introduction

It is well-established that breastfeeding provides numerous benefits to infant development [1, 2]. Milk is a dynamic and living system, and research into milk has provided crucial insight into maternal and infant health. For example, human milk contains developmental and immunological components that optimize neurodevelopment and protects against diseases such as necrotizing enterocolitis [1, 2]. The mechanisms of these beneficial effects are largely unknown, and there is a need for an improved understanding of human milk and its role in infants.

When the goal is to characterize human milk, volunteer donors can ethically and conveniently donate milk to scientific research. Practical and ethical limitations arise, however, for studies evaluating the impact of breastfeeding on the infant. For example, understanding the molecular interactions between milk components and the infant's gastrointestinal tract is

**Data Availability Statement:** Human donor identifying information will be withheld. Other data are available at KiltHub repository, https://doi.org/10.1184/R1/24535045.v1.

**Funding:** Funding was provided to K.W. from the National Institutes of Health (www.nih.gov), award #DP2-HD098860. The funders had no role in study design, data collection and analysis, decision to publish, or preparation of the manuscript.

**Competing interests:** The authors have declared that no competing interests exist.

difficult to impossible, given a dearth of non-invasive externally assessed endpoints. For such analyses, mice are typically the most convenient alternative, and previous studies have used mice to study the roles of milk-derived immune cells [3], bacteria [4], and extracellular vesicles [5, 6].

When performing milk-related studies in mice, researchers have two choices regarding the milk source. The first is to orally administer milk from another species, most typically human or bovine milk [7–9]. Bovine milk is a particularly convenient choice because it is inexpensive and readily available in large quantities; however species differences and differences in milk composition between the milk source and the mouse pup recipient may confound results. The alternative is to work directly with mouse milk to better recapitulate the allogenic milk-infant system present in human breastfeeding studies. Although mouse milk is difficult to collect due to the small size of mice and low volumes of milk available, it is possible, and its use likely provides more robust results than xenogeneic models.

Beyond collection, the other challenge related to mouse milk is that it has not been well-characterized, and thus its appropriateness as a model for human milk remains unclear. Because of this lack of understanding, researchers working with mouse models often extrapolate human milk composition data to mice [10]. This extrapolation is not preferred, however, and thus we were motivated to better characterize mouse milk. Specifically, we are interested in the membrane enclosed structures in milk, which include maternal cells, milk fat globules, and extracellular vesicles [11].

Many early studies assumed that all maternal cells in milk were leukocytes [12]. Since that time, others have directly and indirectly characterized mouse milk cells, with the latter being more common. One method of indirect characterization is examination of milk cell populations recovered from the stomachs of mouse pups [3, 13]. This approach has shortcomings, however, because the majority of cells in the stomach are apoptotic [14]. Another indirect method for characterizing mouse milk is to study the site of milk production in the mouse mammary gland. Although mouse mammary gland biology previously focused on non-lactating and cancerous breast tissue [15, 16], more recent investigations have used single-cell ribonucleic acid (RNA) sequencing to characterize healthy, non-lactating mouse mammary glands [17]. While these indirect approaches to characterizing mouse milk provide useful information, they may not accurately reflect all of the cell populations in milk.

Limited direct characterization of mouse milk cells has been previously conducted. For example, Ikebuchi and colleagues compared immune cells from mouse milk to the cell populations in the mouse mammary gland and reported that T cells are the most common cell type in mouse milk [18]. This differs from human milk, in which T cells represent only 1–2% of the total cell population [19–21]. Unfortunately, this previous study examined only immune cells, which constitute only a subset of maternal milk cells. In humans, most milk cells are mammary epithelial cells, and no characterization has been performed on such cells in mouse milk. Therefore, it is unclear how the cell populations in human and mouse milk compare.

Here, we address this knowledge gap by characterizing cells and extracellular vesicles of mouse milk. Specifically, we examine the density, types, and size distribution of mouse milk cells. Extracellular vesicles are also considered and compared to those present in human milk. Further, we discuss the procedure of mouse milk collection and important considerations during data analysis to assist others in using allogenic mouse milk models. Together, our data offer an improved understanding of the cells in mouse milk and are anticipated to facilitate the use of mouse models, with the ultimate goal of establishing the importance breast milk during early human life.

## Materials and methods

### Animal experiments

Animal protocols were approved by the institutional animal care and use committee at Carnegie Mellon University (Pittsburgh, PA). All animal experiments were conducted in accordance with Protocol PROTO201800009. C57BL/6 mice (Charles River) were housed under controlled temperature (25˚C) in 12-hour light-dark cycles. Animals were given *ad libitum* access to standard diet and water.

### Human milk research

Human milk donors were recruited according to the Institutional Review Board (IRB) protocol number STUDY2019_00000084 at Carnegie Mellon University. Donors met the following inclusion criteria: donors must be over 18 years old, must be breastfeeding a child born in the past 24 months at the time of donation, and must be in general good health. The recruitment period occurred from July 15, 2019 through March 15, 2022. Participants provided informed consent in writing.

### Mouse milk collection and cell isolation

The method for collecting mouse milk was based on the protocol by Willingham et al. [22]. The dam was separated from the pups for two hours prior to milking, while the male mouse kept the pups warm. At the time of milking, the dam was anesthetized using isoflurane and placed on a heating pad, and 2 IU oxytocin was injected intraperitoneally. Oxytocin works quickly to stimulate milk letdown, and within a few minutes of injection, milk letdown begins and milk can be collected. We had the most success in obtaining mouse milk when two people worked together: one person to manually express milk from the mouse, and another person to collect the milk using a 10 μL or 20 μL pipette. We found that small pipette tips collected milk more efficiently than bigger pipette tips, because the capillary forces in the smaller pipette tips help minimize sample loss. The person expressing milk from the mouse used the fingertips of both hands to gently push on the fur around the mouse mammary gland. The person holding the pipette picked up drops of mouse milk as they form on the mouse mammary gland and transferred the milk to a 1.5 mL tube. In our experience, it was easier to collect higher volumes of milk from the mammary glands near the mouse's hind legs.

Mouse milk cells were isolated by centrifugation and washing with PBS. For mouse milk, all centrifugation steps were conducted for 10 minutes at $500 \times g$, 4˚C. First, we combined 100 μL mouse milk with 1 mL PBS in a 1.5 mL tube. This was centrifuged once, and the cell pellet was then transferred to a clean 1.5 mL tube. The cell pellet was then re-suspended in 1 mL fresh PBS and centrifuged a second time. After this second centrifugation step, the resulting cell pellet was re-suspended into 1 mL fresh PBS, and live cell staining for flow cytometry could begin.

### Human milk cell isolation

Because the volumes of human milk were so different from the volumes of mouse milk, the procedures we used for isolating cells from milk were necessarily different. We used the method for isolating cells from human milk previously optimized by Hassiotou et al. [23] Briefly, 20 mL human milk was combined with 20 mL PBS and centrifuged for 20 minutes at $800 \times g$, 4˚C. The cell pellet was re-suspended into 10 mL PBS, transferred to a clean 15mL tube, and centrifuged for 7 minutes at $600 \times g$, 4˚C. The cell pellet was then re-suspended again into 10 mL fresh PBS, and centrifuged again at $600 \times g$, 4˚C. These PBS washing steps

**Table 1. Antibodies used for flow cytometry and Western blotting.**

| Antibody | Fluorophore | Marker | Manufacturer | Cat no. | Clone | Application |
|---|---|---|---|---|---|---|
| CD9 | | Extracellular vesicles | Thermo Fisher | MA5-310980 | | Western |
| CD31 | Brilliant Violet 605 | Endothelial cells | Biolegend | 102427 | | Flow |
| CD45 | APC/Cyanine7 | Immune cells | Biolegend | 103116 | 30-F11 | Flow |
| CD49f | eFluor 450 | Epithelial progenitors | eBioscience | 48-0495-82 | | Flow |
| CD326 (EpCAM) | Brilliant Violet 421 | Epithelial cells | Biolegend | 118225 | G8.8 | Flow |
| Grp94 | | Endoplasmic reticulum | Cell Signaling | 20292S | | Western |
| Nanog | PE | Stem cells | Invitrogen | PA546891 | | Flow |
| TSG101 | | Extracellular vesicles | Thermo Fisher | MA1-23296 | | Western |

help remove fat and protein from the cell pellet; after the second wash, cells were stained for flow cytometry.

## Flow cytometry

The procedure for cell staining is the same for cells isolated from both human and mouse milk. A summary of the antibodies used for flow cytometry is found in Table 1. Live cells from milk were stained using yellow Fixable Viability Dye (FVD) and DRAQ5 (ThermoFisher) as follows: The cell pellet was re-suspended in a 1:1000 dilution of FVD in PBS. The cell suspension was incubated at 37°C in the dark with gentle shaking for 30 minutes. For the last 10 minutes of that incubation time, DRAQ5 was added at a 1:2000 dilution. The cell suspension was then centrifuged for 10 minutes at $500 \times g$, 4°C. The cell pellet was then washed once with PBS to remove free dye before fixing the cells. The cells were then fixed in 4% formaldehyde at room temperature in the dark for 10 minutes, quenched with flow buffer, and centrifuged for 10 minutes at $500 \times g$, 4°C. Antibody staining was conducted as follows: 1 μL antibody was added to 100 μL cell suspension, and staining was conducted on ice in the dark for 30 minutes. After staining, the cell suspension was quenched with flow buffer and centrifuged for 10 minutes at $500 \times g$, 4°C. Flow cytometry was conducted using a Nococyte 3000 flow cytometer (Agilent). Data analysis was conducted using NovoExpress software.

## Milk cell imaging

To image fresh human and mouse milk MESs, 10 μL of fresh milk was placed directly on a slide and covered with a coverslip. Brightfield imaging was conducted on a Keyence BZ-X800 fluorescence microscope. To image cells with staining, the cell pellet was first isolated from a sample of milk as described above. The cell pellet was smeared onto a ColorFrost Plus slide and allowed to air dry overnight. The slide was then washed once with PBS, fixed with 10% formaldehyde in PBS for 10 minutes, and then washed twice with PBS to remove formaldehyde. Following the fixation step, the cells were stained with 1:500 Hoechst and 1:2000 Nile Red (Invitrogen) for 1 hour at room temperature. The slides were then imaged on a Keyence BZ-X800 fluorescence microscope.

## Measuring MES morphology

The morphology of cells and MFGs was measured by flow cytometry by diluting 2 μL of whole milk immediately after expression into 200 μL of PBS, pipetting into a 96-well plate, and running directly on a Novocyte 3000 flow cytometer without any staining or centrifugation steps.

MES size was measured in ImageJ as follows: a threshold was applied to the image, to highlight the MESs and convert the image into black and white. From this thresholding step, the

area of each particle in square microns was calculated in ImageJ. Particles with a size less than one square micron and particles on the edge of the image were excluded. From the area measurements, particle diameter was calculated. For each image, a histogram was generated in GraphPad Prism, with a bin size of 0.5 μm. The histograms of 5–6 images from each milk sample were added together, as shown in S1 Fig.

## Extracellular vesicle isolation from mouse milk

Extracellular vesicles from milk were isolated by a combination of ultracentrifugation and size exclusion chromatography [24–26]. Briefly, 50 μl of fat separated milk was centrifuged at 2,000×$g$ for 10 min at 4˚C and then at 10,000×$g$ for 30 min at 4˚C. The supernatant was then ultracentrifuged at 100,000×$g$ for 3 hours (TL-100 benchtop ultracentrifuge, Beckman-Coulter). The obtained crude EV pellet was washed in PBS once at 100,000×$g$ for 3 hours. The washed pellet was resuspended in 1 ml of PBS and EVs were purified by mini-size exclusion chromatography (mini-SEC) using 1.5 cm x 12 cm mini-columns (Bio-Rad, Hercules, CA, USA; Econo-Pac columns) packed with 10 mL of Sepharose 2B (Millipore-Sigma, St. Louis, MO). Crude EVs (1.0 ml) obtained from ultracentrifugation were loaded onto the column and five 1 ml fractions corresponding to the void volume peak were collected in PBS. Fraction four was collected and used for subsequent experiments as the 'EV' fraction. Given that SEC is a size-dependent assay, we anticipate that the EVs obtained using this approach contain a heterogeneous mixture of exomeres, exosomes, and microvesicles in the size range of 30–200 nm [27]. The EVs were characterized by nanoparticle tracking analysis (NTA) and Western blotting as per MISEV2018 guidelines [28]. Antibodies used for Western blot experiments are described in Table 1.

## TEM imaging

TEM characterization was performed as previously described [29]. Briefly, milk EVs were fixed with 4% glutaraldehyde (Electron Microscopy Services, Hatfield, PA, USA) for 20 min at room temperature (RT). A 10 μL droplet of glutaraldehyde-fixed EVs was placed on Formvar-coated 300 mesh copper grid (Electron Microscopy Services, Hatfield, PA). The sample was incubated for 1 min followed by rinsing with distilled water for 1 min to ensure the removal of PBS salts. Excess liquid was blotted off with a Whatman filter. Post rinsing, 50 μL of the uranyl-acetate solution was put on the grid and allowed to remain for 1 min. Excess liquid was removed, and the grids were viewed on a Hitachi H-7100 transmission electron microscope (Hitachi High Technologies) operating at 100 keV. Digital images were collected using a CCD camera system (AMT Advantage 10, Advanced Microscopy Techniques) and inspected using the NIH ImageJ software.

## Data analysis and statistics

Flow cytometry data analysis was conducted in NovoExpress software. Statistical analysis was conducted using Graphpad Prism 8. Image analysis was conducted using ImageJ. A significant difference was defined as $p < 0.05$.

## Results

### Mouse milk yields vary with litter size

The goal of this study was to characterize mouse milk, as this would enable better comparison to human milk. To that end, we established a procedure to collect milk directly from the mouse mammary gland (Fig 1). During this procedure, the dam was anesthetized two hours

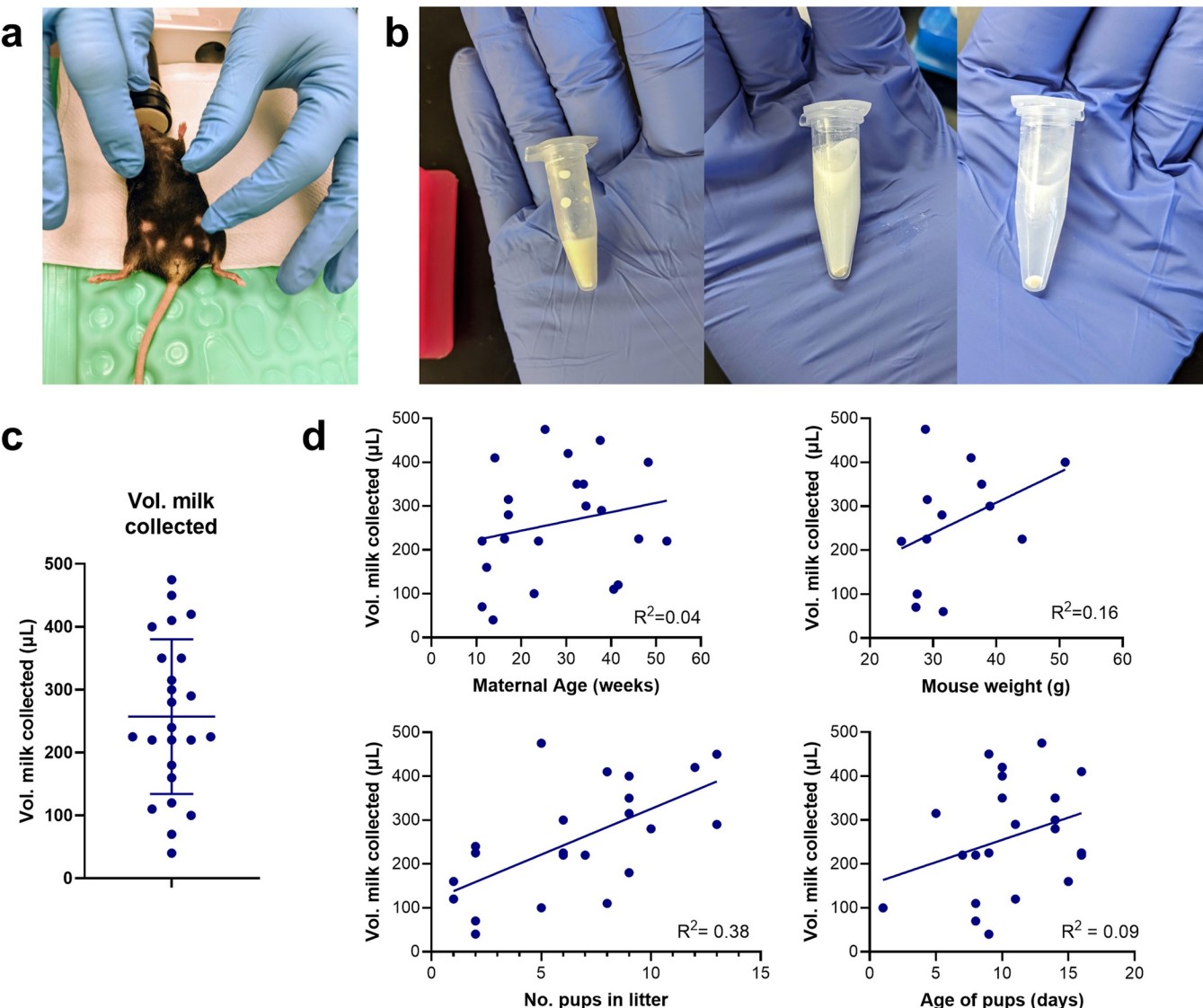

**Fig 1. Overview of the mouse milk collection and cell isolation process.** a) Milk was manually collected from a mouse by separating the dam from the pups for 2 hours, anesthetizing the dam, administering oxytocin, and suctioning the teats with a pipet. b) To isolate mouse milk cells, immediately after collection (left), milk was diluted with phosphate buffered saline (middle) and centrifuged to obtain a cell pellet (right). c) The volume of milk collected from a single mouse ranged from ~50–500 μL. Error bars represent standard deviation. N = 23 mice. d) The strongest predictor of milk yield was the number of pups in the litter. Maternal weight, maternal age, and age of pups did not predict the volume of milk collected from the dam. Each data point represents a single milk collection event.

after separation from the pups and dosed with oxytocin to induce letdown. Milk was collected via pipet by manually expressing the mammary glands (Fig 1A). Isolated cells from the milk were used for further analysis (Fig 1B). The volume of mouse milk collected varied significantly between individuals, with an average volume of 255 μL and a maximum volume of nearly 500 μL (Fig 1C). This compares favorably with previous studies that have collected ~200 μL of milk per dam [30].

At the time of milking, we recorded the age of the dam and the age of pups, the number of pups in the litter, and the weight of the dam. The milk yield correlated with the number of pups in the litter (Fig 1D), which is consistent with previous observations [30]. The milk yield

**Table 2. Characteristics of the mice used in milk collection.**

| Characteristic | Minimum | Maximum | Mean | Median | N |
|---|---|---|---|---|---|
| Mouse weight (g) | 25 | 50.9 | 33.5 | 31.5 | 14 |
| No. pups in litter | 1 | 13 | 6.81 | 7 | 26 |
| Days postpartum | 1 | 16 | 10.38 | 10 | 26 |
| Mouse mother age (weeks) | 11.29 | 52.43 | 27.59 | 25.29 | 23 |
| Vol. milk collected (uL) | 40 | 475 | 242.62 | 225 | 26 |

**Table 3. Correlations of mouse characteristics to volume of milk collected.**

| Characteristic | Pearson r | R squared | P value | Significance | No. XY pairs |
|---|---|---|---|---|---|
| Litter size | 0.6209 | 0.3855 | 0.0012 | ** | 24 |
| Age of pups | 0.2757 | 0.0760 | 0.1922 | ns | 24 |
| Maternal age | 0.2170 | 0.0471 | 0.3321 | ns | 22 |
| Maternal weight | 0.4155 | 0.1727 | 0.1579 | ns | 13 |

did not correlate with the age or weight of the dam, or with the age of the pups. Characteristics of the mice used for milk collection are summarized in Table 2, and statistics are summarized in Table 3.

## Comparing cell density in mouse and human milk

To describe the differences between cells in mouse and human milk, we first compared the total numbers of cells and other cell-like structures in the milk (Fig 2). Quantifying cells in milk is non-trivial, and in recent years, new flow cytometry strategies have improved the accuracy of cell counts in human milk [11, 31]. One reason for the analytical complexity is that milk contains cells and fat globules that are visually indistinguishable without nuclear staining. Milk fat globules (MFGs) are membrane-bound structures varying in size between 0.2 μm and 15 μm that derive from the endoplasmic reticulum of mammary gland epithelial cells [32, 33]. Human and mouse milk fat globules have average diameters between 4–5 and 4–7 μm, respectively [34, 35]. Milk fat globules and milk cells are both present in the cell pellet of milk and are sometimes collectively referred to as membrane enclosed structures, visible by brightfield imaging (Fig 2A) [11].

We asked whether human and mouse milk differ in membrane enclosed structure size distribution and morphology. By measuring the front scatter and side scatter of whole milk by flow cytometry, we found that mouse milk contains more particles with a high front scatter, which correlates to larger size (Fig 2B). To quantify the size distribution, we analyzed the brightfield images of the particles in ImageJ, using the software to calculate the size of the particles and then generating histograms from the resulting data (S1 Fig and Fig 2C). The sizes of the cells and fat globules in milk do not follow a Gaussian distribution. Instead, there are populations of small and large membrane enclosed structures for both humans and mice. For the samples analyzed, human milk contained a higher number of small particles (less than 3μm in diameter) and large particles (greater than 10 μm in diameter) compared to mouse milk. The average human and mouse particle sizes were 4.00 and 4.96 μm, respectively.

To distinguish between cells and milk fat globules in milk, the use of a deoxynucleic acid (DNA) stain is required, as it identifies nucleated cells [11, 31]. In this analysis of human and mouse milk, we used both a DNA stain, DRAQ5, and a membrane stain, yellow Fixable Viability Dye (FVD). DRAQ5 is a DNA stain suitable for staining both live and dead cells in flow

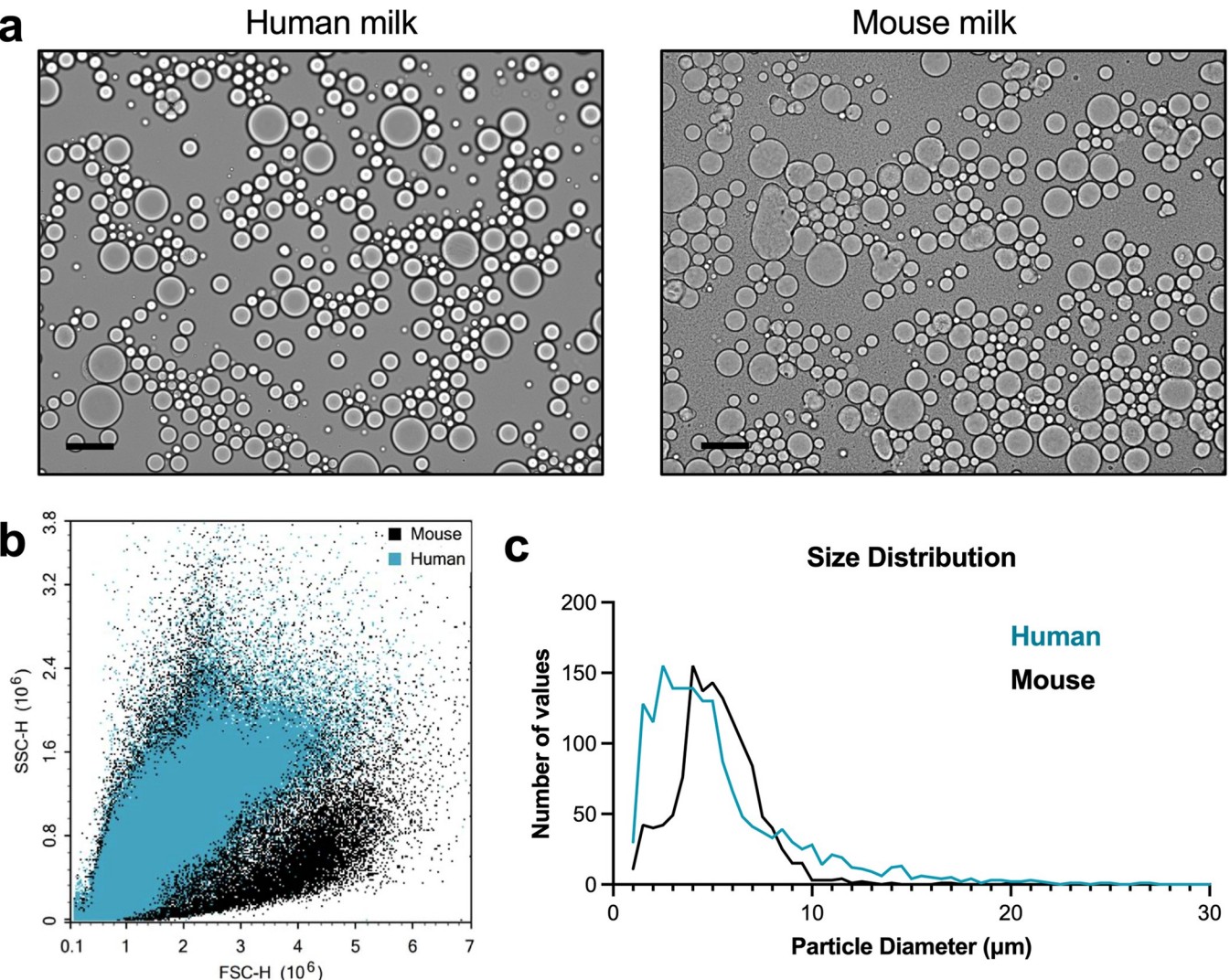

**Fig 2. Mouse and human milk membrane enclosed structures vary in morphology and size distribution.** a) Brightfield images of human and mouse milk immediately after expression show cells and milk fat globules, collectively referred to as membrane enclosed structures. Scale bars: 10 μm. b) Human and mouse milk membrane enclosed structure populations differ in size and granularity as measured by front and side scatter on flow cytometry. c) The size distribution of membrane enclosed structures varies for human and mouse milk samples. This graph was generated from 5–6 images per sample.

cytometry [36]. Fixable viability dye stains cells with compromised membranes and, therefore, causes dead cells to fluoresce [37]. We considered live cells to be DRAQ5$^+$ and FVD$^-$, dead cells to be DRAQ5$^+$ and FVD$^+$, and milk fat globules to be DRAQ5$^-$ and FVD$^+$. Remaining events that were FVD$^-$ and DRAQ5$^-$ included debris and other MFGs. By applying this categorization strategy to mouse cells (Fig 3A) and human cells (Fig 3B), we found that mouse milk contained a higher cell density than human milk by three orders of magnitude (Fig 3C).

Of note, the cell counts in milk were highly variable, with human live cell counts between $10^2$ to $10^5$ cells per mL and mouse cell counts between $10^5$ to $10^7$ cells per mL. In addition to higher average cell counts, mouse milk also had a higher percentage of events that were DRAQ5+, meaning that the mouse milk pellet contains a higher proportion of cells compared to fat globules (Fig 3D). Characteristics of the humans used for milk collection are summarized in Table 4.

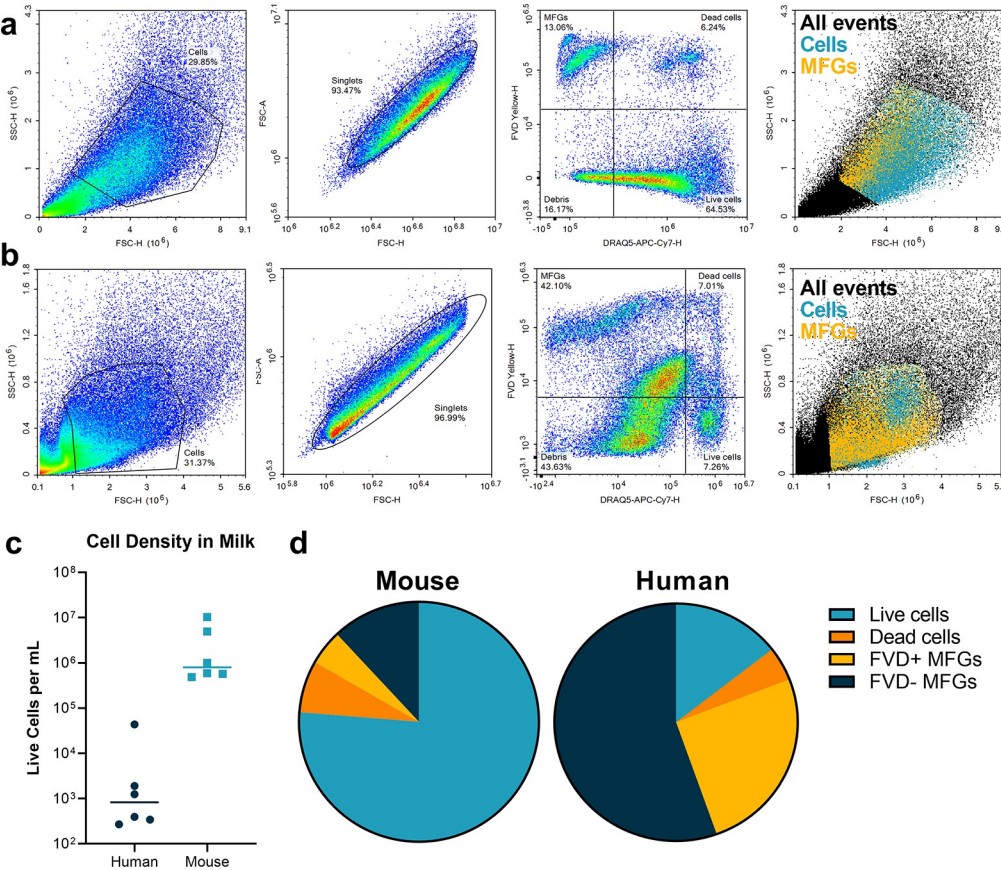

**Fig 3. Mouse milk contains a higher density of cells than human milk.** a) Live and dead cells as well as milk fat globules (MFGs) from mouse milk were identified by flow cytometry using the nuclear stain DRAQ5 and fixable viability dye (FVD). Plots shown are from a 20 µL sample of mouse milk. b) The same process was repeated for live cells from human milk. Flow cytometry plots contain all events from a 10 mL sample of human milk. c) The average cell density of mouse milk is higher than human milk by three orders of magnitude. d) Mouse milk cell pellets contain a much higher proportion of live cells compared to human milk. Pie charts represent the averages from 6 samples of human milk and 7 samples of mouse milk.

## Improving accuracy of cell counts by DNA staining

Next, we used microscopy to confirm our findings from our flow cytometry characterization of milk cells and fat globules (Fig 4). Since we could not image the far-red stain DRAQ5 on our fluorescence microscope, we used Hoechst as a nuclear stain to identify cells. Additionally, we used Nile Red to stain lipid droplets. Since Nile Red has broad excitation and emission spectra, examining Nile Red staining by microscopy gave clearer results than flow cytometry. After staining the pellet from mouse milk with Hoechst and Nile Red, we saw that some membrane enclosed structures in the cell pellet were not nucleated (Fig 4A). These structures, which were Nile Red+ and Hoechst-, were milk fat globules.

These data underscore the importance of using a DNA stain to distinguish nucleated cells from non-nucleated, membrane bound milk fat globules. While using a viability stain excludes dead cells and some fat globules from analysis, its sole use overestimates the number of cells (Fig 4B). This decrease applies to both mouse and human milk, but the difference is more pronounced for human milk (*$p$ = 0.04 using a paired t-test). Fig 4C shows that both DRAQ5 and Hoechst effectively identify nucleated cells in our respective flow and microscopy studies.

**Table 4. Characteristics of human milk donors.**

| Donor no. | Vol. milk (mL) | Week of Lactation | Feeding status | Health status | Flow for cell counts? | EVs? | Imaging? | Time of day |
|---|---|---|---|---|---|---|---|---|
| 1 | 50 | | | Healthy | No | Yes | No | Morning |
| 2 | 50 | 30 | Mixed | | No | Yes | No | |
| 3 | 50 | 21 | | | No | Yes | No | |
| 4 | 160 | 76 | Mixed | Healthy | Yes | No | Yes | Morning |
| 5 | 23 | 32 | Mixed | Healthy | Yes | No | No | Morning |
| 6 | 20 | 26 | Formula | Healthy | Yes | No | Yes | Afternoon |
| 7 | 42 | 24 | EBF | Healthy | Yes | No | No | Morning |
| 8 | 130 | 33 | Mixed | Healthy | Yes | No | Yes | Afternoon |
| 9 | 160 | 31 | Mixed | Infant cold | Yes | No | Yes | Afternoon |

| Donor no. | Notes | Live cells per mL | Total cells per mL | % viability |
|---|---|---|---|---|
| 1 | | - | | |
| 2 | | - | | |
| 3 | | - | | |
| 4 | Whole session | 4.34E+04 | 4.96E+04 | 87.5 |
| 5 | Foremilk | 2.69E+02 | 3.52E+02 | 76.3 |
| 6 | Hindmilk | 3.41E+02 | 5.65E+02 | 60.3 |
| 7 | Hindmilk | 3.94E+02 | 7.44E+02 | 53.0 |
| 8 | Whole session | 1.89E+03 | 2.00E+03 | 94.6 |
| 9 | Whole session | 1.25E+03 | 5.07E+03 | 24.6 |

EBF = exclusively breastfeeding

## Identifying cell types by flow

After quantifying the total number of cells in mouse and human milk, we next characterized the main cell types in mouse milk and compared them to human milk. First, we examined cell morphology to identify cell types visually by imaging mouse and human cells on a fluorescence microscope. The cells from both species contain lipid droplets (Fig 5A). These cells are likely to be lactocytes producing fat, and the presence of lipid droplets within the cells complicates characterization by flow cytometry. Because the cells are highly granular and the milk lipids increase background fluorescence, flow cytometry data is often noisy, and it can be difficult to distinguish positive and negative populations of cells. Fig 5B shows that appropriate controls, such as viability staining and single surface marker staining, can reduce issues with sample noise and facilitate identification of cell types.

By characterizing samples using these controls, we found that mouse milk cell populations were highly variable (Fig 5C). We used CD45 and EpCAM as broad markers for immune cells and epithelial cells, respectively. The presence of epithelial cells and immune cells in mouse milk is consistent with findings from other species, including human milk [21, 38, 39] and cow, goat, and sheep milk [40, 41]. Cell populations in mouse milk as measured by flow cytometry varied between 0 and 100% positive for the markers CD45 and EpCAM. Given the high levels of noise in these samples, it was difficult to obtain precise values for the percentage of immune and epithelial cells in mouse milk, but we can reasonably conclude that immune and epithelial cells are the predominant cell types in mouse milk. Additionally, we analyzed cells for CD49f (mammary epithelial progenitors), CD31 (endothelial cells), and Nanog (stem cells). We found low levels of CD49f and Nanog and no expression of CD31.

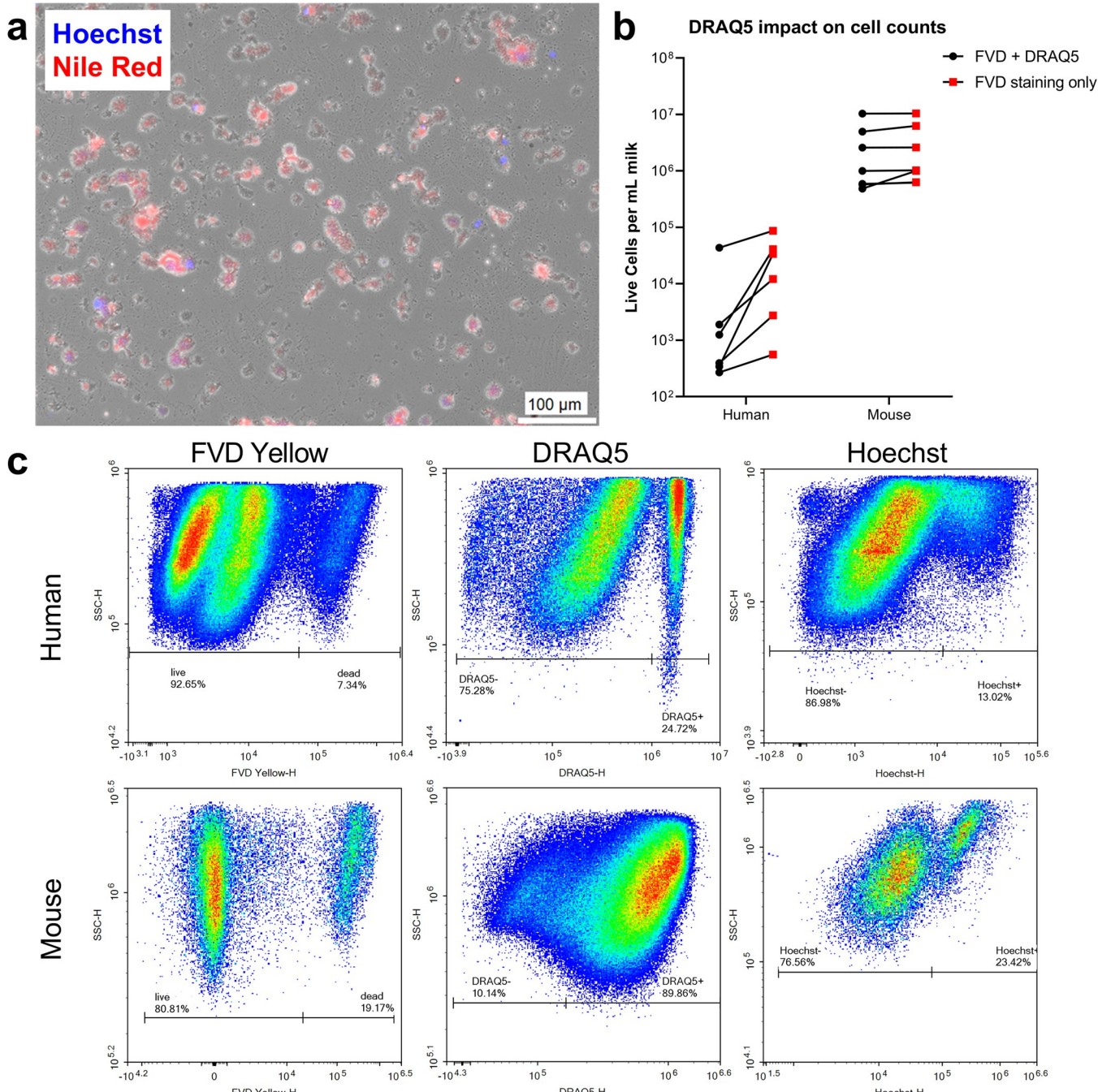

**Fig 4. DNA staining is essential to obtaining accurate cell counts in milk.** a) Hoechst (cell nuclei) and Nile Red (lipids) staining identifies some MESs from mouse milk as nucleated cells (those with overlaid blue and red staining). The remainder of objects are fat globules that can be confused for cells. b) The use of only viability staining when counting cells on a flow cytometer overestimates the number of cells in milk. Staining with both fixable viability dye (FVD) and a DNA stain provides more accurate cell counts. c) Single staining with FVD (to identify particles with compromised membranes), or with a DNA stain such as Hoechst or DRAQ5, is insufficient to identify populations of live cells in milk.

### Extracellular vesicles

In addition to cells, the extracellular vesicles (EVs) in mouse milk have not been well-characterized. Human milk extracellular vesicles were first described in 2007 [42] and have been

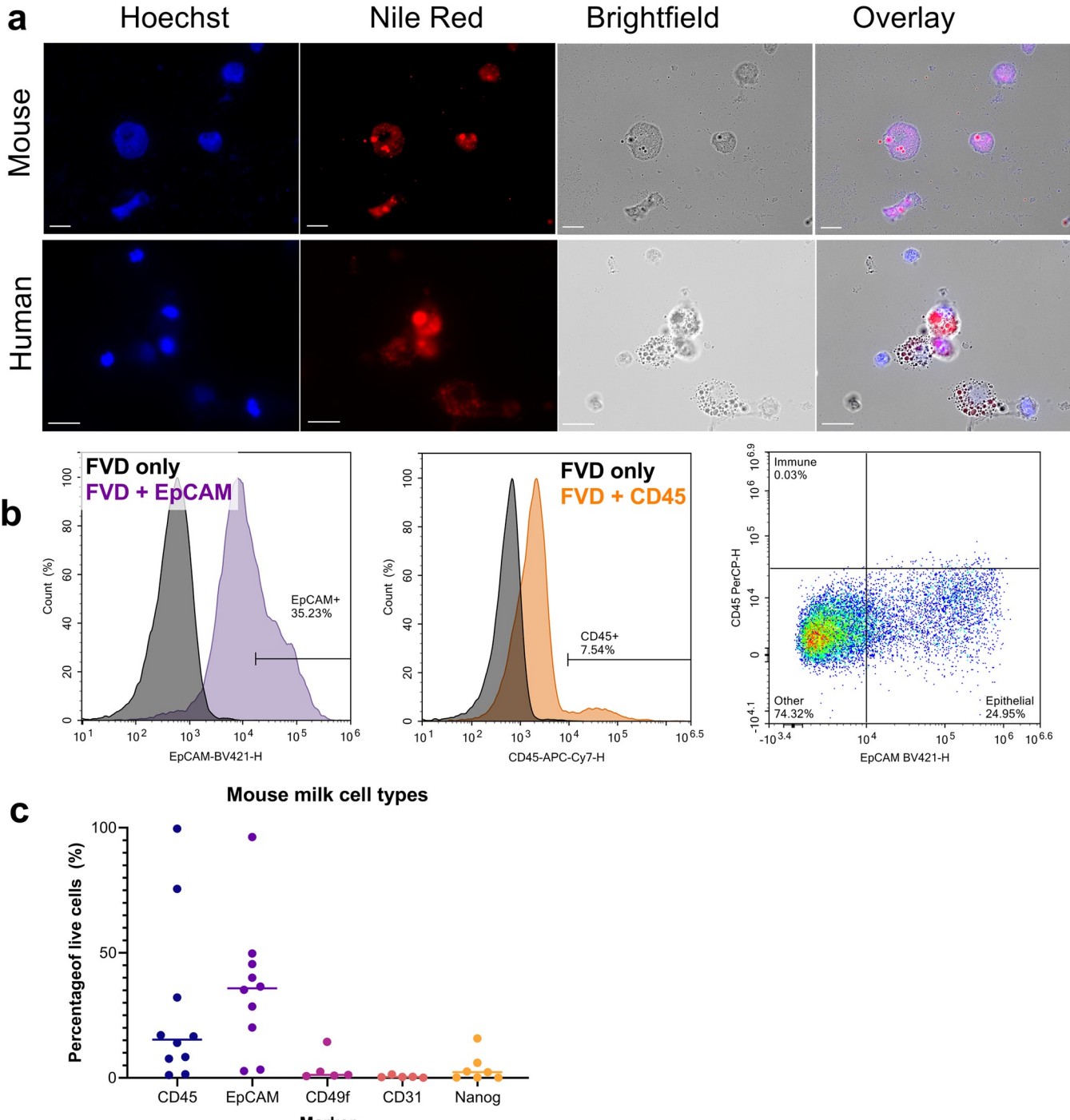

**Fig 5. Mouse milk contains epithelial and immune cells, but cell identification by flow cytometry is highly variable.** a) Staining mouse and human milk cells with Nile Red and Hoechst reveals highly granular cells containing lipid-rich droplets. Scale bar: 20μm. b) Representative flow cytometry plots showing populations of epithelial and immune cells in mouse milk, using the markers EpCAM and CD45, respectively. c) The relative percentages of specific cell populations in mouse milk vary considerably. N = 5–10 samples of mouse milk analyzed for each marker, with each sample coming from mouse milk collected on a different day.

reported to have important roles in cell-to-cell communication [43], as well as intestine health [44] and gene regulation in the infant [45]. Mouse milk extracellular vesicles have not been previously described. To aid their comparison to human milk EVs, we characterized both human and mouse milk EVs by TEM imaging (Fig 6A) and quantified their density and size.

We found that EV density in human milk was approximately an order of magnitude higher than in mouse milk, $\sim 10^{11}$ vs $\sim 10^{10}$ EVs per mL, respectively (Fig 6B). The EVs from the two species were the same size, with mouse and human milk EVs averaging 113 and 127 nm in diameter, respectively (Fig 6C). To confirm that the samples isolated from the mouse milk were pure EVs and not contaminated by other cell components, we used Western blotting and nanoparticle tracking analysis. Western blotting confirmed that the EVs expressed the EV-specific markers CD9 and TGS101, but not the marker from endoplasmic reticulum Grp94 (Fig 6D). This characterization of mouse milk EVs can lead to exciting studies showing the role of the milk EVs in regulating gene expression and carrying cargo to the mouse pup's body.

## Discussion

Human milk and mammary gland composition has been widely studied, particularly given advancements in RNA sequencing technology that have enabled cell characterization with unprecedented detail [19–21, 46]. Much less attention has been paid to the characterization of mouse milk, which is needed if using mouse models to recapitulate human breastfeeding. Here, we describe several important features of mouse milk, including the concentration, size, and types of membrane enclosed structures (e.g. maternal cells, milk fat globules, and extracellular vesicles).

To our knowledge, no other study has measured the cell density of mouse milk using modern flow cytometry methods. The cell counts obtained for human milk in Fig 3 are consistent with other values reported in the recent literature [11]. We were surprised, however, to find that mouse milk cells are approximately three orders of magnitude denser than human milk cells.

However, there are significant structural differences between the mammary glands of humans and mice [47]. Although many genes in the mouse and human mammary gland are conserved at a transcriptomic level [48], it is unclear whether that conservation would extend to the genes in mammary epithelial cell subpopulations in milk. Limited information exists about the populations of cells in mouse milk, and the epithelial cell populations in mouse milk have not been previously described. Given what is known about milk cells in humans and other mammals, we would expect that most cells in mouse milk are lactocytes and immune cells [19, 20]. The high background fluorescence in milk cells complicates fluorescence-based analysis methods such as flow cytometry, and as such, future work might use non-fluorescence-based methods of characterizing milk, such as sequencing technologies, to obtain higher quality data.

We were also interested in the inter-individual variability of mouse milk. Human milk composition varies considerably between individuals and for the same individual depending on time since birth, diet, health status, and time of day [49]. These sources of variability were not present in the mice in this study. All mice were healthy C57BL/6 mice, housed in the same location with the same standard laboratory diet, and all mouse milk was collected at the same time of day. Despite this, mouse milk cell counts ranged from $10^5$ to $10^7$ cells per mL. Likewise, the fraction of cells positive for the markers CD45 and EpCAM as measured by flow cytometry varied widely. These data indicate that milk is highly variable even after controlling for genetics, diet, and environment.

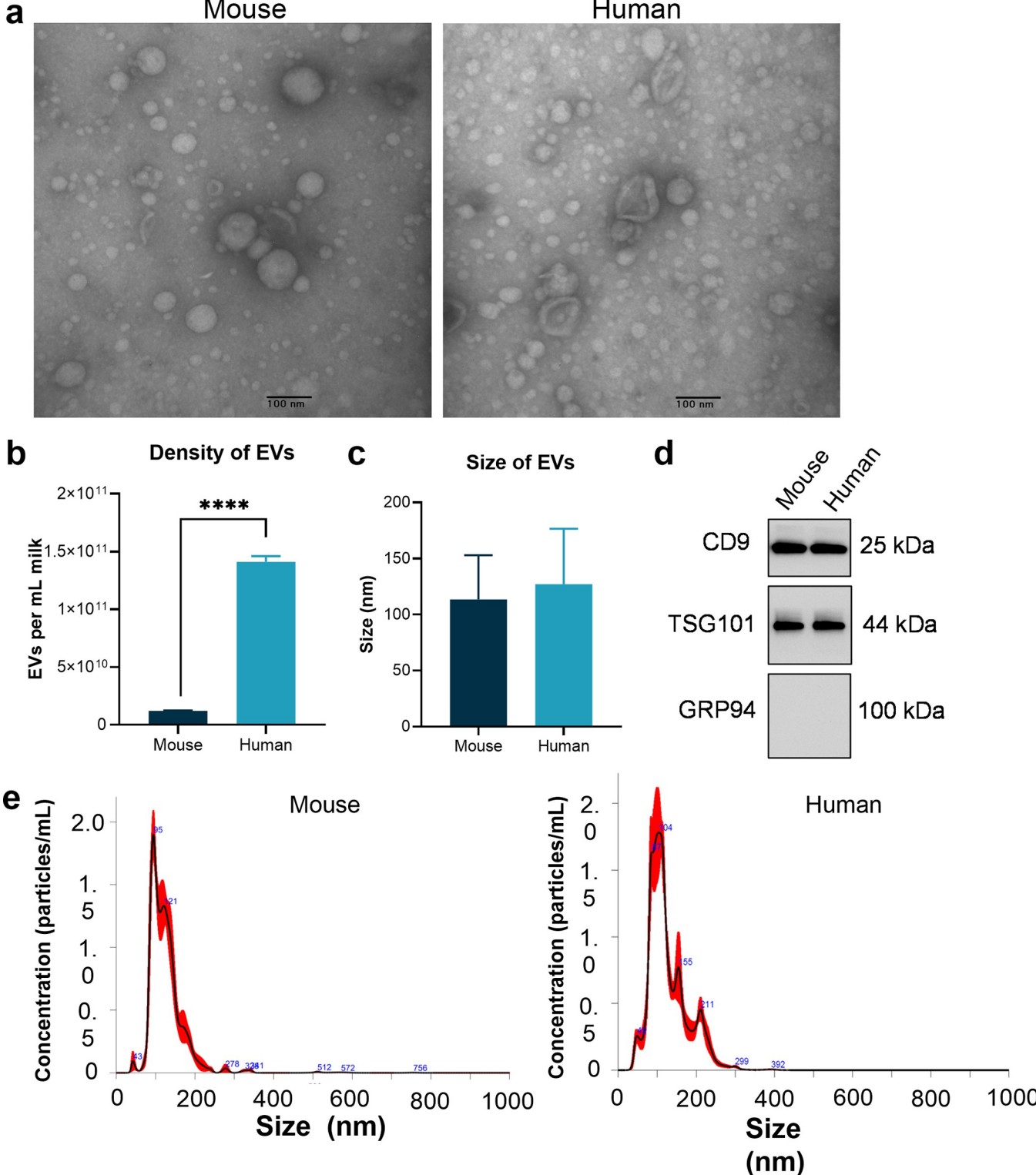

**Fig 6. Mouse milk is rich in extracellular vesicles.** a) Transmission electron microscopy (TEM) images of EVs isolated from mouse and human milk. b) According to nanoparticle tracking analysis, human milk contains an order of magnitude more EVs than mouse milk. c) There is no significant difference in the size of EVs from human and mouse milk as measured by nanoparticle tracking analysis. d) Western blot characterization shows that mouse and human milk EVs express the EV markers CD9 and TGS101, but do not express Grp94, a protein from the endoplasmic reticulum which should not be present in samples of EVs. e) Nanoparticle tracking analysis showing the size distribution of EVs in mouse and human milk.

All species of mammals produce milk, and the basic components of milk–fats, proteins, sugars, cells, nucleic acids–are shared among species. In this research, we have provided an overview of the cells present in mouse milk and outlined some of the major similarities and differences between mouse and human milk. We have also detailed a strategy for accurately isolating, identifying, and quantifying the cells from mouse milk, and outlined some common sources of confusion in milk cell data analysis. Furthermore, we have described extracellular vesicles in mouse milk and compared them to human milk. Milk cells and extracellular vesicles are rapidly developing areas of discovery, and understanding their role using mouse models can shape the future of milk research.

## Conclusions

Here, we developed a method for obtaining substantial quantities of mouse milk and conducted a comparative analysis with human milk to identify similarities and differences. Our findings revealed that both milks consist primarily of epithelial and immune cells. However, significant variations were observed: mouse milk exhibited a cell density three orders of magnitude higher than human milk, while human milk had an EV density that was one order of magnitude greater than mouse milk. These insights inform the limitations of and best utility for the use of mouse models to better understand human milk in the context of infant development.

## Supporting information

**S1 Fig. Representative process of measuring particle size distributions in ImageJ.** For each brightfield image, a threshold was applied in ImageJ to convert the image into black and white. From these black and white images, particle sizes were measured in microns, and a histogram was generated of the particle sizes. These particle size distributions were added together for 5–6 images per sample of milk.
(TIF)

**S1 Raw image.**
(PDF)

## Acknowledgments

The authors would like to thank Angela Malaney and Bethany Fox, who assisted with mouse milk collection, and Alecia-Jane Twigger, who provided helpful discussions and advice. Finally, we would like to thank the human milk donors who contributed milk to these studies.

## Author Contributions

**Conceptualization:** Rose Doerfler, Saigopalakrishna Yerneni, Juliana Hofstatter Azambuja, Kathryn A. Whitehead.

**Data curation:** Rose Doerfler, Saigopalakrishna Yerneni, Kathryn A. Whitehead.

**Formal analysis:** Rose Doerfler, Saigopalakrishna Yerneni, Alexandra Newby.

**Funding acquisition:** Kathryn A. Whitehead.

**Investigation:** Rose Doerfler, Saigopalakrishna Yerneni, Alexandra Newby, Namit Chaudhary, Ashley Shu, Katherine Fein, Juliana Hofstatter Azambuja.

**Methodology:** Rose Doerfler, Saigopalakrishna Yerneni, Alexandra Newby, Namit Chaudhary, Ashley Shu, Katherine Fein, Juliana Hofstatter Azambuja.

**Project administration:** Alexandra Newby, Kathryn A. Whitehead.

**Resources:** Kathryn A. Whitehead.

**Supervision:** Kathryn A. Whitehead.

**Writing – original draft:** Rose Doerfler, Saigopalakrishna Yerneni, Kathryn A. Whitehead.

**Writing – review & editing:** Saigopalakrishna Yerneni, Alexandra Newby, Namit Chaudhary, Ashley Shu, Katherine Fein, Juliana Hofstatter Azambuja, Kathryn A. Whitehead.

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
