## [Decision Letter · Decision Letter 0]

6 Sep 2023

PONE-D-23-22605Characterization and comparison of human and mouse milk cellsPLOS ONE

Dear Dr. Whitehead,

Thank you for submitting your manuscript to PLOS ONE. After careful consideration, we feel that it has merit but does not fully meet PLOS ONE’s publication criteria as it currently stands. Therefore, we invite you to submit a revised version of the manuscript that addresses the points raised during the review process.

We look forward to receiving your revised manuscript.

Kind regards,

Syed M. Faisal, Ph.D.

Academic Editor

PLOS ONE

Journal Requirements:

   "This work was supported by the NIH award #DP2-HD098860. The authors would like to thank Angela Malaney and Bethany Fox, who assisted with mouse milk collection, and Alecia-Jane Twigger, who provided helpful discussions and advice. Finally, we would like to thank the human milk donors who contributed milk to these studies."

   "Funding was provided to K.W. from the National Institutes of Health (www.nih.gov), award #DP2-HD098860. The funders had no role in study design, data collection and analysis, decision to publish, or preparation of the manuscript."

6. Please include a separate caption for each figure in your manuscript.

7. We note you have included a table to which you do not refer in the text of your manuscript. Please ensure that you refer to Table 3 in your text; if accepted, production will need this reference to link the reader to the Table.

8. We notice that your supplementary tables are included in the manuscript file. Please remove them and upload them with the file type 'Supporting Information'. Please ensure that each Supporting Information file has a legend listed in the manuscript after the references list.

9. PLOS ONE now requires that authors provide the original uncropped and unadjusted images underlying all blot or gel results reported in a submission’s figures or Supporting Information files. This policy and the journal’s other requirements for blot/gel reporting and figure preparation are described in detail at https://journals.plos.org/plosone/s/figures#loc-blot-and-gel-reporting-requirements and https://journals.plos.org/plosone/s/figures#loc-preparing-figures-from-image-files. When you submit your revised manuscript, please ensure that your figures adhere fully to these guidelines and provide the original underlying images for all blot or gel data reported in your submission. See the following link for instructions on providing the original image data: https://journals.plos.org/plosone/s/figures#loc-original-images-for-blots-and-gels. 

Additional Editor Comments:

I have examined the feedback provided by our respected reviewers on your manuscript. To progress, it's crucial that you attend to and incorporate all recommended changes and suggested experiments from the reviewers.

Should there be particular suggestions from the reviewers that you find difficult to implement, we kindly request a thorough rationale for that.

We eagerly await your revised manuscript.

Reviewers' comments:

Reviewer's Responses to Questions

**Comments to the Author**

1. Is the manuscript technically sound, and do the data support the conclusions?

Reviewer #1: Yes

Reviewer #2: Partly

Reviewer #3: Yes

2. Has the statistical analysis been performed appropriately and rigorously? 

Reviewer #1: Yes

Reviewer #2: I Don't Know

Reviewer #3: Yes

3. Have the authors made all data underlying the findings in their manuscript fully available?

Reviewer #1: Yes

Reviewer #2: Yes

Reviewer #3: Yes

4. Is the manuscript presented in an intelligible fashion and written in standard English?

Reviewer #1: Yes

Reviewer #2: Yes

Reviewer #3: Yes

5. Review Comments to the Author

Reviewer #1: The manuscript entitled “Characterisation and comparison of human and mouse milk cells” focuses on detailed characterisation of mice milk that includes understanding different type of cells, extracellular vesicles, morphology of these cells. The study compares the similarities and differences of mouse milk and human milk. Also, this study provided a detailed method of isolating milk from mice mammary glands. The study offers an improved understanding of cells in mice milk so that mouse models can be used for studying the importance of breast milk in early stages of human life. The author needs to address some concerns of the reviewer. The manuscript needs minor revision before it can be accepted for publication in the journal. The comments are as follows:

1. The ultimate aim of the authors is to use mice models for studying the importance of breast milk in early stages of human life but as the authors find that there are a lot of differences in the composition, density, percentage of each cell type between mice and human milk; how can the mouse models be appropriate for studying the molecular interactions between milk components and gastrointestinal tracts in human babies?

2. For characterising the milk, the authors have selected the mice with a lot of biological variability in terms of female mouse weight (min 25 and max 50.9) and age (min 11.29 and max 52.43). The authors have no control on the number of litters and volume of milk produced, however the age and weight of mice can be similar to have more consistency in the data. Is there a reason, the authors have chosen mothers of different age and body weights?

Reviewer #2: Introduction: The benefits of breast milk are well described, yet the mechanistic details related to how breast milk protects against acute and chronic diseases and optimizes neurodevelopment remain largely missing in the introduction.

Results:

a. Figure legends are missing the manuscript.

b. The milk yield did not correlate with the age or weight of the dam, or with the age of the pups. However, it seems the numbers of pups/dam is correlated with age and weight of the dam, older/heavier the dam, higher the number of pups, thus the yield of milk is correlated to the age of dam. Please explain.

c. Since Figure legends are not available with MS, it is not possible to ensure that the both images in Fig2a have been taken at same scale/magnification.

d. Authors are suggested to estimate the average cell size by using ImageJ data as demonstrated in recently published research articles. https://doi.org/10.1016/j.bioadv.2022.213205

e. Results in Fig 2c demonstrated highest number of human and mouse milk 2.5 µm and 4.0 µm diameters, respectively. Whereas, in Fig1a micrographs suggested that diameter of human milk cells is larger as compared to mouse milk cells. Please explain. I would suggest you to confirm the cells diameter and elemental composition analyses by SEM and EDAX microscopy, respectively, if possible, because the study is primarily focusing the comparative characterization of two different milk cell types.

f. Fig 3a-b is very blurry. Besides, in absence of figure legends, I cannot find which data reflects the unstained control data.

Reviewer #3: The review article entitled “Characterization and comparison of human and mouse milk cells” is a nice compilation of the human and mouse milk cells expression at different level like size and age etc. and they secret vesicles. This research study is a summary and emphasis of the human and mouse milk. The experimental designing, writing style and the clarity of the exposition are fine, but the conclusions are not clear. I have some comments for the Author:

Comment-1 In introduction section author mentioned that they were interested in Extracellular vesicles, what was the reason and how do you decide these are a extracellular vesicles. I could be something else, If they are vesicle what material they are caring inside these vesicles. Have you ever try to find out the material what they are caring like protein /cells/DNA /RNA etc, If these vesicles are same as already established in the field of different disease then what's a significance of those vesicles.

Comment-2

I know milk is a form of fat and cells, Instead of focusing vesicles you should have to focus on immune cells what kind of specialized cells are secreted from both kind of milk. I was curios to know does the only freshly secreted mouse milk contain maternal cells or they were remain intact in milk in response after sometime or they disappear. You also mentioned milk contain membrane enclosed structures in milk, fat globules, and extracellular vesicles, how it could be possible small vesicles containing all together. Is there any scientific way to prove it. I suggest you should have to specify what kind of specialized vesicles are they. I understand author try to differentiate b/w milk fat globule in both milk with the help of different stain (fig-3) and you found that mouse milk contained a higher cell density than human milk by three orders of magnitude (Fig 3c) I would like to do suggest some minor experiment to check different Immune cells like T /B, NK, Macrophages cells population in milk cells.

I have some suggestion and queries,

The abstract section is not well organized and please modify abstract section.

Author established a procedure to collect milk directly from the mouse mammary gland what kind of procedure you developed, please mentioned briefly in result section for reader purposes.

In line no 144 author mentioned that human and mouse milk differ in membrane enclosed structure, did you try to find out what kind of vesicles they were like exosomes apoptosomes what was that ? Please justify.

In line no 138 Author mentioned that that milk contain cells are prominent with fat globules that are visually indistinguishable without nuclear staining, did you find out what kind of FAT globules were present or from which cell compartment they arise.

In Fig -3 FACS figure can you please change the gating strategy as FSC-H/FSC-A And put a rectangle gate on single cell population with the help of polygonal gate. The round gate is not looking good on single cells population for publication quality purposes.

Fig-4a and in 6a fig resolution is not clear, please change it with good resolution and high brightness level.

The authors are advised to change the FACS gating strategy, try to maintain consistency throughout the FACS DATA.

Conclusion section is poorly written.

Manuscript can be published but it needs standard review with enriches referencing.

Authors are strictly advised to first complete the full form then use abbreviations.

This manuscript provides a topic of interest to the researchers in this field and has a potential for possible publication in this journal after some suggested experiments and minor corrections.

6. PLOS authors have the option to publish the peer review history of their article (what does this mean?). If published, this will include your full peer review and any attached files.

Reviewer #1: **Yes: **Sidra Islam

Reviewer #2: No

Reviewer #3: **Yes: **NIDA MUBIN

---

## [Author Response · Author response to Decision Letter 0]

2 Dec 2023

Reviewer #1: The manuscript entitled “Characterisation and comparison of human and mouse milk cells” focuses on detailed characterisation of mice milk that includes understanding different type of cells, extracellular vesicles, morphology of these cells. The study compares the similarities and differences of mouse milk and human milk. Also, this study provided a detailed method of isolating milk from mice mammary glands. The study offers an improved understanding of cells in mice milk so that mouse models can be used for studying the importance of breast milk in early stages of human life. The author needs to address some concerns of the reviewer. The manuscript needs minor revision before it can be accepted for publication in the journal. The comments are as follows:

1. The ultimate aim of the authors is to use mice models for studying the importance of breast milk in early stages of human life but as the authors find that there are a lot of differences in the composition, density, percentage of each cell type between mice and human milk; how can the mouse models be appropriate for studying the molecular interactions between milk components and gastrointestinal tracts in human babies?

Thank you for your comment. We appreciate your observation, and we concur with the reviewer's point that mouse studies do not always directly correlate with human studies. Despite the differences between mouse and human milk, mouse models facilitate the study of the maternal milk cells within a suckling pup. It is possible to use these models to analyze cell transport within the pup intestine, to study cell interactions with the pup’s immune system, and to check for maternal milk cell uptake into systemic circulation. These experiments are infeasible in a living human baby. So, to make best use of mouse models, we need to understand their limitations, and this is facilitated by the study at hand. As an example, by recognizing similarities in the types of cells present in both human and mouse milk, we can explore cellular interactions in mice. Through this study, we found a higher cell density in mice compared to human milk. This knowledge allows us to infer that the extent of cell-cell interactions in a human infant will be lower than in mice. 

2. For characterising the milk, the authors have selected the mice with a lot of biological variability in terms of female mouse weight (min 25 and max 50.9) and age (min 11.29 and max 52.43). The authors have no control on the number of litters and volume of milk produced, however the age and weight of mice can be similar to have more consistency in the data. Is there a reason, the authors have chosen mothers of different age and body weights?

We deliberately chose different mouse ages and body weights to better represent the diverse population of human mothers. 

--

Reviewer #2: Introduction: The benefits of breast milk are well described, yet the mechanistic details related to how breast milk protects against acute and chronic diseases and optimizes neurodevelopment remain largely missing in the introduction.

We thank you for this comment; we have added additional details in the first paragraph of the Introduction.

Results:

a. Figure legends are missing the manuscript.

We apologize for this oversight. They are now included.

b. The milk yield did not correlate with the age or weight of the dam, or with the age of the pups. However, it seems the numbers of pups/dam is correlated with age and weight of the dam, older/heavier the dam, higher the number of pups, thus the yield of milk is correlated to the age of dam. Please explain.

Thank you for the comment. We plotted volume of milk collected to maternal age, volume of milk collected to mouse weight, volume of milk collected to pups in the litter, and volume of milk collected to age of pups. None of these plots show any strong correlation of mouse characteristics to amount of milk collected. There is no correlation with age of the dam to volume of milk correlated. The R2 value is 0.04, which is <<1.

c. Since Figure legends are not available with MS, it is not possible to ensure that the both images in Fig2a have been taken at same scale/magnification.

We have included the scale bar information in the legend, and the bars are identical for both images. 

d. Authors are suggested to estimate the average cell size by using ImageJ data as demonstrated in recently published research articles. https://doi.org/10.1016/j.bioadv.2022.213205

Thank you for the suggestion. The average size in human milk is 4.00 μm, and in mouse milk it's 4.96 μm. We have now included this data in the results section of the manuscript. The method published by Ali et. al. in the article mentioned is identical to what we used in the original manuscript, with the only difference being the way it is plotted.

e. Results in Fig 2c demonstrated highest number of human and mouse milk 2.5 µm and 4.0 µm diameters, respectively. Whereas, in Fig1a micrographs suggested that diameter of human milk cells is larger as compared to mouse milk cells. Please explain. I would suggest you to confirm the cells diameter and elemental composition analyses by SEM and EDAX microscopy, respectively, if possible, because the study is primarily focusing the comparative characterization of two different milk cell types.

Thank you for this comment. We have replaced Fig 2A to be more visually representative of the distribution. The originally images were also correct, however. In those original images, we see a wider spread in size for human milk compared to mouse milk. Our eyes are naturally drawn to the larger particles, but if you look more closely, there are many more smaller particles. The human (blue) line in Fig 2C is wider in both overall shape but also in the width of the peak compared to the black line—showing a wider diversity in sizes. The larger particles in human milk correspond to the tail of Fig 2C, where it is clear there are more particles between 10-20 μm in human milk than mouse. Also note that the for the size distribution graph, 5-6 images were used to develop this, which we have included in the caption. 

SEM and EDXA microscopy are techniques are primarily used in material science to investigate material structure and composition. We do not anticipate they will provide additional insights given that SEM and analyzing EDAX microscopy data will be challenging due to the heterogeneous biological nature of fat globules.

f. Fig 3a-b is very blurry. Besides, in absence of figure legends, I cannot find which data reflects the unstained control data.

Thank you for your comment. For some reason, the images that appear in the PDF version that reviewers receive are not high quality. Please click the hyperlink in the document to download the TIF files of the figure, which are at the resolution required by the journal.

--

Reviewer #3: The review article entitled “Characterization and comparison of human and mouse milk cells” is a nice compilation of the human and mouse milk cells expression at different level like size and age etc. and they secret vesicles. This research study is a summary and emphasis of the human and mouse milk. The experimental designing, writing style and the clarity of the exposition are fine, but the conclusions are not clear. I have some comments for the Author:

Comment-1 In introduction section author mentioned that they were interested in Extracellular vesicles, what was the reason and how do you decide these are a extracellular vesicles. I could be something else, If they are vesicle what material they are caring inside these vesicles. Have you ever try to find out the material what they are caring like protein /cells/DNA /RNA etc, If these vesicles are same as already established in the field of different disease then what's a significance of those vesicles.

Thank you for your comment. Extracellular vesicles are membrane-bound structures secreted by both eukaryotic and prokaryotic cells. In the past decade, an expanding body of literature has demonstrated the pivotal role of EVs in cell-to-cell communication, impacting normal physiology and disease pathology. These vesicles are also present in biological fluids like milk, blood, urine, carrying diverse cargo, including proteins, nucleic acids, sugars, and metabolites.

Recent reports have highlighted that the cargo within milk EVs can exert local (gut) and systemic effects in infants and adults who consume them. Due to this increasing interest in milk EV biology, we embarked on a study of murine and human milk EVs. In order to characterize these isolated EVs, we adhered to the isolation and characterization guidelines provided by the International Society for Extracellular Vesicles (Théry et al., J Extracell Vesicles, 2018). Our characterization process included evaluating their size (NTA), structure (TEM), and examining the presence of known protein cargo (western blot).

Comment-2

I know milk is a form of fat and cells, Instead of focusing vesicles you should have to focus on immune cells what kind of specialized cells are secreted from both kind of milk. I was curios to know does the only freshly secreted mouse milk contain maternal cells or they were remain intact in milk in response after sometime or they disappear. You also mentioned milk contain membrane enclosed structures in milk, fat globules, and extracellular vesicles, how it could be possible small vesicles containing all together. Is there any scientific way to prove it. I suggest you should have to specify what kind of specialized vesicles are they. I understand author try to differentiate b/w milk fat globule in both milk with the help of different stain (fig-3) and you found that mouse milk contained a higher cell density than human milk by three orders of magnitude (Fig 3c) I would like to do suggest some minor experiment to check different Immune cells like T /B, NK, Macrophages cells population in milk cells.

Thank you for your comment. We agree that examining immune and other cell populations in milk is a significant aspect. Our group (Gleezon et al., Science Advances 2022) and others (e.g., Twigger et al., Nature Communications 2022) have previously reported on the cell populations in human milk. 

Regarding extracellular vesicles, they encompass various subtypes distinguished by their size and molecular content. Historically, vesicles falling within the size range of 20-250 nm were commonly labeled as exosomes. However, in recent years, the International Society for Extracellular Vesicles (ISEV) has adopted a more inclusive term, "small extracellular vesicles" (sEV), for vesicles within this size range. This change is due to the current imperfections in available methods and protocols for isolating different extracellular vesicle subtypes. Hence, we have chosen to use 'small extracellular vesicles' rather than exosomes.

I have some suggestion and queries,

The abstract section is not well organized and please modify abstract section.

We are unclear why the reviewer feels the abstract section is not well-organized. Our abstract structure conforms to most writing guides (please see this Nature guide). The abstract provides a basic introduction to the field and more detailed background information to motivate the main problem that the manuscript addresses. Then, we describe the results and put it in broader perspective.

Author established a procedure to collect milk directly from the mouse mammary gland what kind of procedure you developed, please mentioned briefly in result section for reader purposes.

We thank you for this comment. You can find this information in the Fig. 1 caption now, and we have also added an abbreviated description to the Results section as requested. 

In line no 144 author mentioned that human and mouse milk differ in membrane enclosed structure, did you try to find out what kind of vesicles they were like exosomes apoptosomes what was that ? Please justify.

 Thank you for your comment. The membrane-bound structures we are discussing are indeed exosomes, but we refer to them as 'small extracellular vesicles' (sEVs), which is the current convention per the International Society for Extracellular Vesicles. This choice is due to the practical difficulty in obtaining a completely 'pure' subset of EVs. In Figure 6, we characterized the sEVs by their size (utilizing Nano tracking analysis), structure (examined through Transmission electron microscopy), and the presence of protein cargo (detected using western blotting for exosome-specific markers). 

In line no 138 Author mentioned that that milk contain cells are prominent with fat globules that are visually indistinguishable without nuclear staining, did you find out what kind of FAT globules were present or from which cell compartment they arise.

Thank you for this suggestion. The kind of fat globules in milk has been established previously as milk fat globule (MFG) (Nakatani et al., J Biochem, 2013; Chai et al., Food Sci Animal Resour, 2022). This is stated in approximately line 180. It is known that these come from the mammary gland epithelia (Lee et al., Frontiers in Pediatrics, 2018). The components and compositions of the MFG are also well established. MFGs originate as small, triacylglycerol-rich, droplets that are formed on or in endoplasmic reticulum membranes. These droplets are released from endoplasmic reticulum into the cytosol as microlipid droplets coated by proteins and polar lipids. Microlipid droplets can fuse with each other to form larger cytoplasmic lipid droplets (Lee et al., Frontiers in Pediatrics, 2018).

We have added details to the text with appropriate references. 

In Fig -3 FACS figure can you please change the gating strategy as FSC-H/FSC-A And put a rectangle gate on single cell population with the help of polygonal gate. The round gate is not looking good on single cells population for publication quality purposes.

Thank you for your comment and your concern that gating be done correctly. Flow cytometry analysis can be approached in various ways, and both our way and the strategy you suggest will yield the same findings, as the current oval gate encompasses all the cells that would be considered within the polygonal gate.

Fig-4a and in 6a fig resolution is not clear, please change it with good resolution and high brightness level.

Thank you for your comment. For some reason, the images that appear in the PDF version that reviewers receive are not high quality. Please click the hyperlink in the document to download the TIF files of the figure, which are at the resolution required by the journal.

The authors are advised to change the FACS gating strategy, try to maintain consistency throughout the FACS DATA.

Thank you for the comment. Flow cytometry analysis offers flexibility in approach, and maintaining a consistent gating strategy for all samples is not always feasible and/ accurate (Lonati et al., Scientific Reports 2021; Van Epps, Immunity & Ageing, 2014). Nevertheless, we are committed to transparency regarding our chosen gating strategy, which is why we show the gating. This ensures that readers have a comprehensive understanding of our data analysis.

Conclusion section is poorly written.

Thank you for bringing this to our attention. We were missing a Conclusion section and have now included one. 

“Here, we developed a method for obtaining substantial quantities of mouse milk and conducted a comparative analysis with human milk to identify similarities and differences. Our findings revealed that both milks consist primarily of epithelial and immune cells. However, significant variations were observed: mouse milk exhibited a cell density three orders of magnitude higher than human milk, while human milk had an EV density that was one order of magnitude greater than mouse milk. These insights inform the limitations of and best utility for the use of mouse models to better understand human milk in the context of infant development.”

Manuscript can be published but it needs standard review with enriches referencing.

Thank you for your suggestion. We have added references in response to the reviewers’ comments.

Authors are strictly advised to first complete the full form then use abbreviations.

We have ensured abbreviations are used only after the full form was stated.

This manuscript provides a topic of interest to the researchers in this field and has a potential for possible publication in this journal after some suggested experiments and minor corrections.

Thank you.

---

## [Decision Letter · Decision Letter 1]

15 Jan 2024

Characterization and comparison of human and mouse milk cells

PONE-D-23-22605R1

Dear Dr. Whitehead,

We’re pleased to inform you that your manuscript has been judged scientifically suitable for publication and will be formally accepted for publication once it meets all outstanding technical requirements.

Kind regards,

Syed M. Faisal, Ph.D.

Academic Editor

PLOS ONE

Additional Editor Comments (optional):

Reviewers' comments:

Reviewer's Responses to Questions

**Comments to the Author**

1. If the authors have adequately addressed your comments raised in a previous round of review and you feel that this manuscript is now acceptable for publication, you may indicate that here to bypass the “Comments to the Author” section, enter your conflict of interest statement in the “Confidential to Editor” section, and submit your "Accept" recommendation.

Reviewer #1: All comments have been addressed

Reviewer #3: All comments have been addressed

2. Is the manuscript technically sound, and do the data support the conclusions?

Reviewer #1: Yes

Reviewer #3: Yes

3. Has the statistical analysis been performed appropriately and rigorously? 

Reviewer #1: Yes

Reviewer #3: Yes

4. Have the authors made all data underlying the findings in their manuscript fully available?

Reviewer #1: Yes

Reviewer #3: Yes

5. Is the manuscript presented in an intelligible fashion and written in standard English?

Reviewer #1: Yes

Reviewer #3: Yes

6. Review Comments to the Author

Reviewer #1: The manuscript entitled “Characterization and comparison of human and mouse milk cells” addressed all the points asked in the first revision and needs no further revision. The manuscript can be accepted in present form.

Reviewer #3: I agree with the authors comments. The author described a comment and fulfil the criteria very well.

7. PLOS authors have the option to publish the peer review history of their article (what does this mean?). If published, this will include your full peer review and any attached files.

Reviewer #1: **Yes: **Sidra Islam

Reviewer #3: **Yes: **NIDA MUBIN

---

## [Editor Report · Acceptance letter]

21 Jan 2024

PONE-D-23-22605R1 

PLOS ONE

Dear Dr. Whitehead, 

I'm pleased to inform you that your manuscript has been deemed suitable for publication in PLOS ONE. Congratulations! Your manuscript is now being handed over to our production team.

Kind regards, 

on behalf of

Dr. Syed M. Faisal 

Academic Editor

PLOS ONE